# 12-month clinical outcomes of combined phacoemulsification and ab interno trabeculectomy for open-angle glaucoma in the United Kingdom

Ejaz Ansari[1]*, Deva Loganathan[2]

1 Maidstone & Tunbridge Wells NHS Trust and University of Kent, Canterbury, United Kingdom, 2 Maidstone & Tunbridge Wells NHS Trust, Maidstone, United Kingdom

* e.ansari@nhs.net

## Abstract

### Background/Objectives

To describe intraocular pressure (IOP) and ocular hypotensive medication outcomes of combined phacoemulsification and ab interno trabeculectomy with the Kahook Dual Blade (KDB; New World Medical, Inc, Rancho Cucamonga, CA) in adults with cataract and open-angle glaucoma (OAG).

### Subjects/Methods

Retrospective chart review of existing medical records. Data collected included intraocular pressure (IOP) and IOP-lowering medication use preoperatively and through up to 24 months postoperatively. Paired t-tests were utilized to compare preoperative to postoperative mean IOP and mean medications used.

### Results

Data from 32 eyes of 26 subjects were analyzed. Subjects were predominantly Caucasian (25/26) had mean (standard error) age of 79.3 (1.2) years, and eyes had moderate-advanced OAG (mean visual field mean deviation -8.3 [1.3] dB). Mean IOP was 19.8 (0.8) mmHg at baseline and 15.5 (0.6) mmHg (p<0.0001) after mean follow-up of 11.5 (1.0) months; IOP reductions of ≥20% were achieved in 20/32 eyes (62.5%). Mean medication use declined from 2.4 (0.2) medications per eye at baseline to 0.5 (0.2) at last follow-up (p<0.0001); 23/32 eyes (71.9%) were medication-free at last follow-up. No vision-threatening complications were observed.

### Conclusions

Combined phacoemulsification and ab interno trabeculectomy with the KDB safely provided mean IOP reductions of 21.7% and mean IOP medication reductions of 83% after mean follow-up of 12 months in eyes with moderate to advanced OAG. This procedure provides

**Data Availability Statement:** All relevant data are within the paper.

**Funding:** The author(s) received no specific funding for this work.

**Competing interests:** The authors have declared
that no competing interests exist.

medication-independence in most eyes with statistically and clinically significant IOP
reductions.

## Introduction

Glaucoma is the leading cause of irreversible blindness worldwide [1], and the prevalence of
primary open-angle glaucoma (POAG) among adults over age 65 years in England and Wales
—the most common form of glaucoma in the region [2]—is 3% [3] and in Northern Ireland is
2.83% [4]. Vision loss or blindness from glaucoma represents approximately 7–15% of all
vision loss/blindness in the United Kingdom (UK) [5–7], resulting in direct health care costs
of approximately £200 million annually [5].

Primary therapeutic options for POAG include both medical and laser therapy options,
both of which effectively lower intraocular pressure (IOP) and prevent glaucoma-related vision
loss [8]. Historically, surgical therapies such as ab externo trabeculectomy or tube-shunt
implantation have been reserved for more severe or recalcitrant cases. More recently, the
development of numerous minimally invasive surgical options that lower IOP somewhat less
than older procedures but with more favorable safety profiles has expanded indications for
surgery earlier in the disease course and in the treatment cascade [9–11]. Many of these proce-
dures can be performed either as standalone surgery in eyes with inadequate IOP control or in
combination with cataract surgery to reduce IOP and/or the medication burden in eyes with
coincident glaucoma.

One such procedure is an ab interno trabeculectomy. In this procedure, a strip of trabecular
meshwork (TM) is removed from the anterior chamber angle of the eye using a specialized
instrument—the Kahook Dual Blade (KDB, New World Medical, Rancho Cucamonga, CA)—
to facilitate aqueous humor drainage into Schlemm's canal [12]. Ab interno trabeculectomy
can be performed as a standalone procedure or in combination with phacoemulsification cata-
ract surgery (KDB-phaco). In previous studies conducted primary in the United States, KDB-
phaco produced IOP reductions ranging from 12–27% and medication reductions of 21–71%
[13–22].

In this paper, we present clinical outcomes of KDB-phaco through up to 24 months of fol-
low-up in an academic practice in the United Kingdom.

## Subjects and methods

This was a retrospective analysis of existing data drawn from the medical records of qualifying
patients. The protocol for data collection and analysis was reviewed and approved by the
Maidstone and Tunbridge Wells NHS Trust Research and Development Committee, which
granted a waiver of consent.

Subjects included in this analysis were adults aged 18 years or older with visually significant
cataract and medically-controlled open-angle glaucoma who underwent KDB-phaco in one or
both eyes at the Maidstone and Tunbridge Wells Hospitals, England, between September 2017
and May 2019. Patients with glaucomas other than open-angle glaucoma, and those with cor-
neal pathology precluding accurate assessment of IOP by applanation tonometry, were
excluded. Prior incisional glaucoma surgery (prior iridotomy was acceptable if angles were
open), presence of vitreous in the anterior chamber, presence of intraocular silicone oil, clini-
cally significant inflammation or infection in the study eye within 30 days prior to the preoper-
ative visit, active ophthalmic disease/disorder that could confound study results and impaired

episcleral venous drainage were also excluded. All qualifying patients within this time frame were included to preclude selection bias.

The procedure has been described previously [12, 13]. Briefly, under topical tetracaine 1% and intracameral lidocaine 1% anesthesia and following standard phacoemulsification and intraocular lens implantation (EyeCee One, Bausch & Lomb UK, Kingston-Upon-Thames, England), the KDB was introduced into the anterior chamber through the temporal clear corneal surgical incision and advanced to the nasal angle. The TM was engaged with the instrument's tip, which was advanced until the instrument's heel was seated within Schlemm's canal. The instrument was then advanced along Schlemm's canal for the intended extent of TM excision; during advancement, TM tissue was stretched and elevated up the integrated ramp to two parallel blades that excised a strip of TM, which was removed with forceps or with the KDB itself. At the completion of surgery, all ocular hypotensive therapy was discontinued, and dexamethasone 0.1%, nepafenac 0.1%, and pilocarpine 2% each dosed x4 daily for 4 weeks were prescribed. Data retrieved from electronic health records included demographic and baseline glaucoma status data, visual acuity and IOP and medication data at baseline and every postoperative visit, and adverse events occurring intra- or postoperatively. Data were drawn from the preoperative assessment visit, the surgical visit, and the follow-up visits closest in time to 2 weeks, 6 weeks, 3–6 months, and the last follow-up visit for patients followed beyond 4 months and up to 24 months.

The primary outcomes were reductions from baseline in IOP and IOP-lowering medications at each postoperative time point. These were assessed using paired t-tests, with p = 0.05 taken as the level of significance. Visual acuity (VA) changes from baseline were similarly assessed using the logMAR form of VA. Safety assessment included the nature, frequency, and timing of adverse events. As no pre-specified hypotheses were being tested, *a priori* power and sample size calculations were not undertaken, and the sample size was set arbitrarily by the number of cases available for analysis at the time of data acquisition.

## Results

This analysis includes data from 32 eyes of 26 subjects undergoing combined KDB-phacoemulsification. Demographic and baseline glaucoma status data are given in Table 1. Patients

**Table 1. Demographics and baseline glaucoma status in 32 eyes of 26 subjects.**

| Parameter | Value |
|---|---|
| Subject-Level (n = 26) | |
| Age (yr), mean (SE) | 79.3 (1.4) |
| Gender, n (%) | |
| Male | 10 (38.4) |
| Female | 16 (61.6) |
| Ethnicity, n (%) | |
| Caucasian | 25 (96.1) |
| Asian | 1 (3.9) |
| Eye-Level (n = 32) | |
| Operative eye, n (%) | |
| Right | 16 (50) |
| Left | 16 (50) |
| Visual field mean deviation (dB), mean (SE) | -8.3 (1.3) |
| Cup-disc ratio, mean (SE) | 0.75 (0.0) |
| Follow-up, mean (SE) [range] (months) | 11.9 (1.0) [4–24] |

**Table 2. Intraocular pressure, medication, and visual acuity data at each time point.**

|  | Baseline | Week 2 | Week 6 | Month 3–6 | Last Follow-Up |
|---|---|---|---|---|---|
| Number of Eyes | 32 | 12 | 19 | 32 | 32 |
| Intraocular pressure |  |  |  |  |  |
| Mean (SE), mmHg | 19.8 (0.8) | 17.9 (0.9) | 15.1 (1.0) | 15.6 (0.7) | 15.5 (0.6) |
| Mean (SE) change from baseline, mmHg | --- | -2.8 (1.5) | -4.0 (0.9) | -4.3 (0.9) | -4.3 (0.8) |
| Mean (SE) % change from baseline, % | --- | -10.6 (5.8) | -20 (4.5) | -18.7 (4.2) | -18.9 (3.9) |
| p (mean change from baseline) | --- | 0.0945 | 0.0002 | < .0001 | < .0001 |
| Medications |  |  |  |  |  |
| Mean (SE), n | 2.4 (0.2) | --- | --- | 0.1 (0.0) | 0.5 (0.2) |
| Mean (SE) change from baseline, n | --- | --- | --- | -2.3 (0.2) | -1.9 (0.2) |
| Mean (SE) % change from baseline, % | --- | --- | --- | -98.3 (1.2) | -82.5 (6.4) |
| P (mean change from baseline) | --- | --- | --- | < .0001 | < .0001 |
| Best-corrected visual acuity |  |  |  |  |  |
| Mean (SE), logMAR | 0.52 (0.08) | 0.25 (0.07) | 0.11 (0.24) | 0.11 (0.04) | 0.09 (0.04) |
| Mean (SE) change from baseline, logMAR | --- | -0.08 (0.08) | -0.47 (0.25) | -0.41 (0.09) | -0.42 (0.07) |
| Mean (SE) % change from baseline, % | --- | -46.8 (24.4) | -74.1 (50.6) | -67.6 (12.3) | -84 (6.2) |
| P (mean change from baseline) | --- | 0.3504 | 0.0675 | < .0001 | < .0001 |

logMAR, logarithm of the minimum angle of resolution; mmHg, millimeters of Mercury; SE, standard error

were Caucasian (96.2%, 25/26), were on average 79.3 (1.4) years of age, and had moderate or advanced OAG (mean visual field mean deviation -8.3 [1.3] dB). Mean follow-up was 11.5 (1.0) months (range 6–24 months).

Mean IOP data at each time point is given in Table 2. Mean IOP was 19.8 (0.8) mmHg at baseline; significant IOP reductions were seen from Week 6 through the end of follow-up, ranging from 4.1–4.3 mmHg (p≤0.0002). At last follow-up, IOP reductions of ≥20% were achieved in 20/32 eyes (62.5%). Final IOP ≤18 mmHg was achieved in 84.4% of eyes (27/32) and IOP ≤ 15 mmHg was achieved in 56.3% of eyes (18/32) (Table 3).

Mean medication use at each time point following stabilization of the medication regimen at Month 3 is also given in Table 2. The mean number of medications used per eye was 2.4 (0.2) at baseline, by Month 3–6 was reduced to 0.1 (0.0) (p<0.0001), and at last follow-up was 0.5 (0.2), an 82.5% reduction (p<0.0001). At last follow-up, the medication burden has been reduced by at least 1 medication in 87.5% of eyes (28/32), and 71.9% (23/32) were medication-free.

The procedure was well tolerated by all patients. Four eyes (12.5%) had transient hyphema either intraoperatively or within the first few postoperative days, all of which resolved spontaneously. No eyes required any additional glaucoma procedures during the observed follow-up

**Table 3. Pre-specified IOP and medication outcomes at last follow-up.**

| Number of eyes (n) | 32 |
|---|---|
| Proportion achieving IOP reduction ≥20% compared to baseline, n(%) | 20 (62.5) |
| Proportion achieving IOP ≤18 mmHg, n (%) | 27 (84.4) |
| Proportion achieving IOP ≤15 mmHg, n(%) | 18 (56.3) |
| Proportion using ≥ 1 fewer medication compared to baseline, n(%) | 28 (87.5) |
| Proportion medication-free, n (%) | 23 (71.9) |

IOP, intraocular pressure; mmHg, millimeters of Mercury

period. Mean logMAR BCVA improved from 0.52 (0.8) to 0.09 (0.04) (p<0.0001); BCVA was improved (90.6% [29/32]) or unchanged (within 1 line; 9.4% [3/32]) in all eyes, with no eyes losing >1 lines of BCVA.

## Discussion

This study demonstrates that KDB-phaco, performed in eyes with both cataract and open-angle glaucoma, effectively lowers both IOP and the need for IOP-lowering medications in adult patients in the UK. The majority (72%) of these eyes—many with moderate or advanced glaucoma—were medication-free with IOP reductions of 20% or more (62.5%) at last follow-up (6–24 months postoperatively). The procedure was safe, with minimal and self-limited complications that were not sight threatening.

The IOP and medication reductions seen in this study (21.7% and 83%, respectively) are consistent with those reported in other populations. In studies of 6–12 months' duration conducted largely in the US in eyes with predominantly OAG, KDB-phaco provided mean IOP reductions 12–27% and medication reductions of 21–71% [13–22]. Less is known about outcomes of KDB-phaco in international populations. A recently-reported retrospective study from Saudi Arabia of 10 standalone and 40 KDB-phaco cases found mean IOP reduction of 29% and mean medication reduction of 86% 4–7 months postoperatively [23]. In a mixed sample of subjects from the US, Mexico, and Switzerland, all with severe or refractory glaucoma, mean IOP reductions of 24% and medication reductions of 37% were reported at 6 months [24]. Overall, these international studies in patients of various ethnicities demonstrate consistency in the efficacy of the KDB-phaco procedure. In this study, reflective of the authors' practice, effort was made to reduce medications whenever possible postoperatively while still maintaining target IOP, in an effort to improve patients' quality of life and minimize ocular surface symptoms associated with topical ocular hypotensive therapy as a means of sparing the conjunctiva for possible future bleb-based surgery [25–27]. Had medications been discontinued less determinedly, IOP reductions may have been greater at the expense of medication reductions. Variations among investigators of prior studies in discontinuing medications postoperatively may explain in part the substantial inter-study variability in IOP and medication reductions reported.

Glaucoma and cataract both increase in prevalence with increasing age and frequently coexist. Elective cataract surgery affords an opportunity for the surgical redress of glaucoma, however until only recently the surgical options for glaucoma at the time of cataract surgery have been limited to full-thickness procedures—trabeculectomy and tube-shunt implantation—with safety profiles that limited their value as add-on procedures to cataract surgery in patients with moderate IOP or medication reduction goals. In recent years, a growing family of less invasive procedures has been developed for patients in whom modest IOP reductions, or reductions in the medication burden, are desirable [9–11]. These relatively safer procedures are generally less traumatic to ocular tissues and offer faster visual rehabilitation compared to trabeculectomy or tube-shunt implantation [28] and can thus be paired with cataract surgery for co-management of cataract and glaucoma. These procedures are generally classified into those that shunt aqueous humor into Schlemm's canal, the suprachoroidal space, or the sub-conjunctival space, and have been extensively reviewed elsewhere [9–11, 28–32]. As the current study and previous studies discussed above demonstrate, the KDB-phaco procedure effectively reduces both IOP and the IOP-lowering medication burden at the time of cataract surgery while providing excellent visual rehabilitation. Significantly, the KDB-phaco procedure does not induce unexpected postoperative refractive errors [33].

This study's key strength is its inclusion of a sample representing a population in which results of KDB-phaco have not been previously reported, broadening the evidence base for this procedure's international value in addressing coexisting cataract and glaucoma. The length of follow-up—through up to 24 months—is also a strength given the chronicity of glaucoma and the relatively shorter duration (6–12 months) of most prior studies of ab interno trabeculectomy [13–22]. Its retrospective design is a limitation, although the risk of selection bias was minimized by including all consecutive eyes undergoing the procedure during the included time frame. This is also an uncontrolled study, and cataract surgery alone can lower both IOP and the medication burden, although the reductions expected from phacoemulsification alone —14% IOP reduction and 0.5 medication reduction at 1 year [34]—are not sufficient to account for the efficacy outcomes observed in our series.

In summary, the combination of phacoemulsification and ab interno trabeculectomy with the Kahook Dual Blade significantly lowered both IOP and the need for IOP-lowering medications, with excellent visual acuity rehabilitation, in patients with cataract and glaucoma in the United Kingdom.

## Acknowledgments

Assistance with manuscript preparation was provided by Tony Realini, MD, MPH, with support from New World Medical.

## Author Contributions

**Conceptualization:** Ejaz Ansari.

**Data curation:** Deva Loganathan.

**Investigation:** Ejaz Ansari, Deva Loganathan.

**Methodology:** Ejaz Ansari.

**Supervision:** Ejaz Ansari.

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
