## [Decision Letter · Decision Letter 0]

6 May 2021

PONE-D-21-11179

PLOS 1

12-Month Clinical Outcomes of Combined Phacoemulsification and Ab Interno Trabeculectomy for Open-Angle Glaucoma in the United Kingdom

PLOS ONE

Dear Dr. Ansari,

Thank you for submitting your manuscript to PLOS ONE. After careful consideration, we feel that it has merit but does not fully meet PLOS ONE’s publication criteria as it currently stands. Therefore, we invite you to submit a revised version of the manuscript that addresses the points raised during the review process.

More details are needed in the methods section about how were patients included in the study. In addition, the authors should emphasize that not all patients completed a 1 year follow up

We look forward to receiving your revised manuscript.

Kind regards,

Ahmed Awadein, MD, Ph.D, FRCS

Academic Editor

PLOS ONE

Journal Requirements:

2. Thank you for including your ethics statement: "The protocol for data collection and analysis was reviewed and approved by the Hospital Research and Development Committee, which granted a waiver of consent."

3. Please note that according to our submission guidelines (http://journals.plos.org/plosone/s/submission-guidelines), outmoded terms and potentially stigmatizing labels should be changed to more current, acceptable terminology. To this effect, please change "Caucasian" to "White" or "of European descent" (as appropriate).

Reviewers' comments:

Reviewer's Responses to Questions

**Comments to the Author**

1. Is the manuscript technically sound, and do the data support the conclusions?

Reviewer #1: Partly

Reviewer #2: Yes

2. Has the statistical analysis been performed appropriately and rigorously? 

Reviewer #1: I Don't Know

Reviewer #2: Yes

3. Have the authors made all data underlying the findings in their manuscript fully available?

Reviewer #1: Yes

Reviewer #2: Yes

4. Is the manuscript presented in an intelligible fashion and written in standard English?

Reviewer #1: Yes

Reviewer #2: Yes

5. Review Comments to the Author

Reviewer #1: The authors look at the results of combined phacoemulsification with ab-interno trabeculectomy using the Kahook Dual blade. Although the topic is interesting I don't think it adds much to previously published studies on this subject. The authors state that one of the key strengths of the study is the long duration of follow-up, which despite being 1 year on average, ranges from as short as 4 months (Table 1) or 6 months (line 83). It would be useful to compare the results of combined cases with phaco alone in patients with moderate-severe glaucoma. and over a longer follow-up period.

Reviewer #2: In this interesting manuscript, the author evaluated 12-Month Clinical Outcomes of Combined Phacoemulsification and Ab Interno Trabeculectomy for Open-Angle Glaucoma in the United Kingdom and concluded that combined phacoemulsification and ab interno trabeculectomy with the KDB safely provided mean IOP reductions of 19% and mean IOP medication reductions of 83% after mean follow-up of 12 months in eyes with moderate to advanced OAG.

the author is kindly requested to comment on:

1- the inclusion and exclusion criteria.

2- a short description of the surgical technique, anaesthesia, IOL type.

6. PLOS authors have the option to publish the peer review history of their article (what does this mean?). If published, this will include your full peer review and any attached files.

Reviewer #1: No

Reviewer #2: No

---

## [Author Response · Author response to Decision Letter 0]

17 May 2021

Editor-in-Chief

PLOS 1

14th May 2021

Dear Sir/ Madam

Re: PONE-D-21-11179 PLOS 1 12-Month Clinical Outcomes of Combined Phacoemulsification and Ab Interno Trabeculectomy for Open-Angle Glaucoma in the United Kingdom

Many thanks for the useful and constructive comments by the reviewers. 

Response to Reviewers:

Response to Reviewer 1: 

Variable follow-up of this cohort precludes any absolute descriptor of the follow-up period. We elected to include the mean follow-up in the title to provide readers with context. The mean is the standard summary statistic for continuously- and normally-distributed data. As for a phaco-only control group, we would be unable to collect such data as our practice is to perform phaco-MIGS in eyes such as these to reduce IOP and/or the medication burden. Thus, eyes undergoing phaco alone would be sufficiently different from study eyes as to introduce confounding factors. As for comparisons to eyes with more advanced glaucoma, we typically perform alternative procedures in those eyes.

Response to Reviewer 2: 

1- the inclusion and exclusion criteria- these have been included in the text (see tracked)

2- a short description of the surgical technique, anaesthesia, IOL type- these have been included (see tracked).

Thank you in advance for your time and consideration.

Sincerely,

Ejaz Ansari, MD, for the research team

---

## [Editor Report · Decision Letter 1]

24 May 2021

PLOS 1

12-Month Clinical Outcomes of Combined Phacoemulsification and Ab Interno Trabeculectomy for Open-Angle Glaucoma in the United Kingdom

PONE-D-21-11179R1

Dear Dr. Ansari,

We’re pleased to inform you that your manuscript has been judged scientifically suitable for publication and will be formally accepted for publication once it meets all outstanding technical requirements.

Kind regards,

Ahmed Awadein, MD, Ph.D, FRCS

Academic Editor

PLOS ONE
---

## [Editor Report · Acceptance letter]

9 Jun 2021

PONE-D-21-11179R1 

12-Month Clinical Outcomes of Combined Phacoemulsification and Ab Interno Trabeculectomy for Open-Angle Glaucoma in the United Kingdom 

Dear Dr. Ansari:

I'm pleased to inform you that your manuscript has been deemed suitable for publication in PLOS ONE. Congratulations! Your manuscript is now with our production department. 

Kind regards, 

on behalf of

Dr. Ahmed Awadein 

Academic Editor

PLOS ONE